# Evaluation of Febrile Neutropenia in Hospitalized Patients with Neoplasia Undergoing Chemotherapy

**DOI:** 10.3390/microorganisms11102547

**Published:** 2023-10-12

**Authors:** Maria Bachlitzanaki, George Aletras, Eirini Bachlitzanaki, Ippokratis Messaritakis, Stergos Koukias, Asimina Koulouridi, Emmanouil Bachlitzanakis, Eleni Kaloeidi, Elena Vakonaki, Emmanouil Kontopodis, Nikolaos Androulakis, Georgios Chamilos, Dimitrios Mavroudis, Petros Ioannou, Diamantis Kofteridis

**Affiliations:** 1Department of Internal Medicine, Venizeleion General Hospital of Heraklion, 71409 Heraklion, Greece; 2School of Medicine, University of Crete, 71003 Heraklion, Greece; 3Department of Cardiology, Venizeleion General Hospital of Heraklion, 71409 Heraklion, Greece; 4Laboratory of Translational Oncology, Medical School, University of Crete, 70013 Heraklion, Greece; 5Department of Medical Oncology, University General Hospital of Heraklion, 70013 Heraklion, Greece; 6Department of Surgery, Venizeleion General Hospital of Heraklion, 71409 Heraklion, Greece; 7Laboratory of Toxicology, Department of Anatomy, School of Medicine, University of Crete, 70013 Heraklion, Greece; 8Department Medical Oncology, Venizeleion General Hospital of Heraklion, 71409 Heraklion, Greece; 9Department of Clinical Microbiology and Microbial Pathogenesis, School of Medicine, University of Crete, 71003 Heraklion, Greece; 10Department of Internal Medicine and Infectious Diseases, University Hospital of Heraklion, 71110 Heraklion, Greece

**Keywords:** febrile neutropenia, solid tumors, hospitalization, duration of neutropenia, neutropenia

## Abstract

Febrile neutropenia (FN) is a common but serious complication encountered in patients with cancer and is associated with significant morbidity and mortality. In this prospective study, 63 patients with solid tumors under chemotherapy or immunotherapy were admitted to the hospital due to febrile neutropenia, confirmed through clinical or microbiological documentation. The aim of this study was to provide a comprehensive overview of the epidemiological and microbiological characteristics of hospitalized neutropenic patients with solid tumors undergoing treatment. Additionally, we aimed to assess the duration of neutropenia and identify factors influencing patient outcomes. The median age of patients was 71 ± 10.2 years, most of which were males (66.7%), and the primitive tumor location was the lung (38.1%), with most patients (82.5%) being at disease stage IV. The median duration of neutropenia was three days (range 1–10), and, notably, mucositis was significantly associated with neutropenia lasting ≥3 days (*p* = 0.012). Patients with lung cancer (38.1%) and patients with stage IV disease (82.5%) presented a higher risk of FN, although these differences did not reach statistical significance. The site of infection was identifiable in 55.6% of patients, with positive cultures detected in 34.9% and positive blood cultures (BC) drawn in 17.5% of cases. Gram-positive bacteria were the predominant causative agents in BC (63.6%), with *Staphylococci* being the most prevalent among them (66.7%). The median duration of hospitalization was nine days (range, 3–43 days), and most patients showed improvement or cure of infection (16.9% and 74.6%, respectively). Among recorded risk factors, the Eastern Cooperative Oncology Group (ECOG) performance status (PS) appears to be statistically significant. Patients with an impaired PS score (2–4) experienced worse outcomes and higher likelihood of mortality (*p* = 0.004). Regarding the outcome, a longer duration of neutropenia was also statistically significant (*p* = 0.050). Of the patients, 12.7% ultimately succumbed to their conditions, with 37.5% attributed to infections. FN is a common yet serious complication in solid tumor patients. Adequate knowledge of the predictors of mortality and the microbiological causes are of utmost importance to allow accurate diagnosis and prompt treatment as they significantly influence patient outcomes.

## 1. Introduction

Febrile neutropenia (FN) is a common and severe complication in cancer patients, with occurrences in 10–50% of solid tumor patients and over 80% of hematologic malignancy patients [1,2]. This risk is particularly pronounced in individuals undergoing cytotoxic therapies during chemotherapy [3]. Incidence rates vary based on patient-related risk factors, tumor type, treatment modality, and genetic susceptibility factors, such as GSTP1, UGT1A1, MDM2 SNP309, and TP53 R72P genotypes [4,5,6,7]. High-risk groups continue to experience elevated rates of serious complications (25–30%) and mortality (9–12%) [8].

In 65% of FN episodes among solid tumor patients, a clinical focus can be identified [9]. However, microbiological documentation is possible in only 20–30% of cases [10,11], and positive blood cultures (BC) are found in merely 10–25% of patients [12,13]. Gram-negative bacteremia, often attributed to *Pseudomonas aeruginosa*, was the primary causative agent. Yet, recent decades have witnessed shifts in microbiology trends. For instance, there has been a notable increase in Gram-positive bacteria, with an approximate 3:2 ratio [14,15]. Factors contributing to the rise in Gram-positive cocci infections include the more frequent use of intravenous-access devices [16], prophylaxis with quinolones, and aggressive systemic chemotherapy accompanied by severe oropharyngeal mucositis of grade 3/4 [17]. These selective pressures predominantly impact hematologic patients and have a lesser impact on solid tumor patients [9,11].

Common Gram-positive bacteria found in cancer neutropenic patients include *Staphylococcus*, *Streptococcus*, and *Enterococcus* species. In contrast, drug-resistant organisms encompass *Pseudomonas aeruginosa*, *Acinetobacter* species, *Stenotrophomonas maltophilia*, *Escherichia coli,* and *Klebsiella* species [17]. Infections caused by anaerobic microorganisms and polymicrobial infections appear to be relatively uncommon, typically occurring in specific situations such as abscesses or enteritis. In recent years, there has been an increase in strains resistant to extended-spectrum β-lactamase (ESBL) or carbapenemases [18]. This rise in resistant microorganisms is influenced by factors such as comorbidities, previous colonization, recent invasive procedures, prior hospitalization, and previous antimicrobial treatment, as well as the local pattern of resistances [17,19,20,21].

Infections primarily have a bacterial etiology, although the possibility of a viral or fungal origin exists [17]. In the context of invasive fungal infections, they are believed to be uncommon within this patient population [22]. Such infections may be linked to prior antimicrobial usage, multiple chemotherapy lines, high-dose steroid administration (prednisone doses exceeding 20 mg/day for four weeks or longer), the presence of central venous catheters (CVC), or mucositis or prolonged neutropenia (lasting more than seven days) [9]. *Candida albicans* stands as the most frequent cause of candidemia. Nonetheless, there has been a recent surge in the isolation of non-albicans *Candida* spp. among solid malignancy patients, with the incidence of each species varying according to institutional and geographic region [23]. This increase has been particularly noticeable in candidemias caused by fluconazole-resistant species (e.g., C. krusei and C. glabrata). Despite the availability of an expanded array of antifungal treatments, the 30-day mortality for immunocompromised patients with candidemia remains between 31.7% and 39% [9,22].

Prolonged hospitalization and antimicrobial treatment can lead to a reduction in treatment intensity or generate delays in cancer treatment, potentially compromising the efficacy of chemotherapy [24]. Consequently, apart from its pharmacoeconomic implications [25], neutropenia has a detrimental effect on overall survival and the quality of life [26,27]. In certain circumstances, primary prophylaxis for FN becomes essential, especially for the most aggressive antineoplastic regimens or for more vulnerable patient subgroups, such as older adults [27,28].

The primary objective of the current study was to provide an epidemiological and microbiological profile of hospitalized neutropenic solid tumor patients undergoing chemotherapy or immunotherapy. Additionally, we aimed to assess the duration of neutropenia and identify the factors that influence patient outcome.

## 2. Materials and Methods

### 2.1. Patients Sample

A prospective study encompassing all FN episodes confirmed through clinical or microbiological documentation in 63 solid tumor patients was conducted. These patients were administered intravenous or per os chemotherapy or immunotherapy while hospitalized in the oncology departments of the two largest tertiary hospitals of the island of Crete, Greece—the University General Hospital of Heraklion and the Venizeleion General Hospital of Heraklion. The observation period spanned from January 2019 to December 2021.

### 2.2. Definition of Neutropenia and Neutropenic Fever

Neutropenia was defined as an absolute neutrophil count (ANC) falling below 1500 neutrophils/mm^3^, with its severity categorized as follows: mild (ANC 1000–1500 neutrophils/mm^3^), moderate (500–1000 neutrophils/mm^3^), severe (200–500 neutrophils/mm^3^ or ANC expected to decrease below 500 neutrophils/mm^3^ in the next two days), and very severe (<200 neutrophils/mm^3^) or “agranulocytosis” [29].

Neutropenic fever was characterized as a single oral temperature exceeding 38.3 °C (101 °F) or a temperature higher than 38 °C (100.4 °F) sustained for at least one hour, with an accompanied ANC < 1500 neutrophils/mm^3^ [30].

### 2.3. Inclusion Criteria

The study enrolled patients with active solid malignancies undergoing intravenous or per os chemotherapy or immunotherapy who were diagnosed with FN either clinically or through microbiological infection confirmation. FN was diagnosed either upon patient admission or during hospitalization.

### 2.4. Exclusion Criteria

Patients with hematologic malignancies were excluded from the study. Moreover, patients who did not meet the diagnostic criteria for FN, as defined below, or those who had FN episodes resulting from causes other than cancer (e.g., drug-induced FN, primary immunodeficiency, or liver disease) were also excluded.

### 2.5. Data Collection

Clinical and epidemiological characteristics of the patients were recorded, including age, sex, and comorbidities, as well as type and stage of malignancy, current cancer therapy, duration of hospitalization, presence and type of fever, severity and duration of neutropenia, use of granulocyte colony-stimulating factor (GCSF), culture results, and the antimicrobials administered and outcome. When indicated, blood, urine, sputum, stool, and pus samples were collected and cultured using conventional methods and potential pathogens were identified through routine bacteriological procedures.

### 2.6. Statistical Analysis

Statistical analysis was performed using the SPSS 26.0 statistical software package (SPSS Inc., Chicago, IL, USA) and data were presented in percentages. Descriptive statistics, including median, distribution range, and percentage values, were employed to analyze the distribution characteristics of all data. Categorical variables were expressed as n (%). To ascertain risk factors for the duration of neutropenia and its outcome, binary logistic regression analysis was employed. Associations between categorical variables were explored using the Chi-square and Fisher exact tests. Correlations between continuous variables were assessed using the Spearman or Pearson method, based on the variable’s adherence to a normal distribution. A *p*-value < 0.05 was considered statistically significant.

### 2.7. Study Approval

This study was approved by the scientific committee of the University General Hospital of Heraklion, Crete (Protocol number: 74/3/11-3-2020), the scientific committee of the Venizeleion General Hospital of Heraklion, Crete (Protocol number: 4/1/29-01-2020), and the Ethics committee of the University of Crete (Protocol number: 9644/15-1-2020). It adheres to the principles of the Helsinki Declaration.

## 3. Results

The median age of the patients was 71 ± 10.2 years. Forty-two (66.7%) of them were males. The primitive tumor location was lung (24 out of 63, 38.1%), and more specifically, 14 patients (22.2%) were diagnosed with small cell lung cancer (SCLC) and 10 (15.9%) with non-small cell lung cancer (NSCLC), followed by colorectal (9.5%), and pancreatic tumors (7.9%). Most patients (82.5%) were of stage IV. Considering patients’ responses, 57.1% were of stable disease (SD), 33.3% of progressive disease (PD), and 9.5% of partial response (PR). The characteristics of the enrolled patients diagnosed with FN are summarized in Table 1.

Concerning hospitalization etiology, fever and infection were the most common reason (63.5% and 27%, respectively), while 63.5% mentioned no previous hospitalization within the last three months. Regarding therapy, 96.8% were receiving chemotherapy, 1.6% immunotherapy, and 1.6% a combination of chemotherapy and immunotherapy. Six patients (9.5%) were also undergoing radiotherapy. Except for therapy, other risk factors were evaluated, including no G-CSF prophylaxis (15.9%), impaired Eastern Cooperative Oncology Group (ECOG) performance status (PS) 2–4 (27%), or cardiovascular comorbidities (39.7%). Oropharyngeal mucositis was reported in only 11.1% of patients, and considering drug use, 42.9% of patients had received antimicrobial therapy within the last trimester, whereas 11.1% received corticosteroids in the previous month. Seventeen patients (27%) had had surgery or minor invasive procedures within the last thirty days (Table 1).

The median duration of neutropenia in patients included in the study was three days (range, 1–10), and 53 out of 63 patients were treated with prophylactic G-CSF before the febrile episode. In total, 25 (39.7%) had a central venous catheter (CVC), and 7 out of 63 (11.1%) had another type of foreign body. The duration of neutropenia was related to patients’ characteristics, comorbidities, therapy administered, and overall outcome (Table 1).

A statistical analysis revealed mucositis significantly associated with neutropenia ≥3 days (*p* = 0.012). Considering the outcome, a longer duration of neutropenia was also statistically significant (*p* = 0.050).

All patients presented with a fever over 38.3 °C with a median duration of two days (range, 1–7), and only five patients (7.9%) appeared with a recurrent fever of a median time of three days (range 1–10). Absolute neutrophil count and duration of neutropenia were also evaluated. Of all FN episodes, 33.3% of patients had an absolute neutrophil count of <500/mm^3^, 39.7% of patients had 500–1000/mm^3^, and 27% of them had 1000–1500 mm^3^. A detailed description of the type and grade of neutropenia and lymphopenia or thrombocytopenia is demonstrated in Table 2, according to Common Terminology Criteria for Adverse Events (CTCAE) version 5.0, published in November 2017 by National Institutes of Health, National Cancer Institute.

Collected cultures included blood, urine, stool, pus, and sputum samples; in total, 39 positive cultures were detected. The site of the culture and pathogens isolated are summarized in Table 3.

Positive cultures were detected in 22 out of 63 (34.9%), and positive blood cultures (BC) in 11 out of 63 (17.5%).

In our cohort, Gram-positive bacteria constituted the predominant causative agents in BC (63.6%), which can likely be attributed to the high rate of CVC usage among our patients, accounting for 25 out of 63 (39.7%). Among patients with CVC, 8 out of 25 (32%) exhibited detected pathogens in their BC, with Gram-positive pathogens predominating (75%), specifically *Staphylococci* (66.7%). Noteworthy, Gram-negative bacteria included *Escherichia coli* (25.6%), *Enterobacter cloacae* (12.8%), and *Pseudomonas aeruginosa* (7.7%).

Two out of sixty-three patients (3.2%) presented fungal infections with positive cultures. This low incidence is likely attributed to the fact that most patients experienced neutropenia for less than three days. In the case of the first patient, *C. parapsilosis* was isolated from urine culture. Conversely, the second patient’s blood and urine cultures revealed *E. cloacae*, and their neutropenia lasted for one day, ultimately resulting in improvement. The second patient, on the other hand, had positive blood and sputum cultures for *C. tropicalis* and also suffered from a polymicrobial infection, which included methicillin-resistant *S. hominis* from BC and pan-drug resistant *A. baumannii* from a urine culture. This patient endured neutropenia for seven days, and the outcome was fatal.

In 10 out of 63 patients (15.9%), antimicrobial therapy was modified or intensified due to various reasons, including culture results (2 out of 10, 20%), disease worsening in 5 out of 10 patients (50%), recurrence of fever (2 out of 10, 20%), or an allergic reaction (1 out of 10, 10%).

As indicated in Table 4, the site of infection remained unidentified in 28 cases (44.4%). The median duration of hospitalization was nine days (range 3–43), and the majority of patients demonstrated improvement or cure of infection (47 out of 63, 74.6%, and 8 out of 63, 12.7%, respectively). Among the eight patients (12.7%) who did not survive, 37.5% succumbed to the infection, and 62.5% passed away due to disease progression or other causes.

Considering the recorded risk factors, as outlined in Table 4, PS (ECOG) appears to be statistically significant. Patients with impaired PS scores (2–4) exhibit worse outcomes and a higher incidence of death (*p* = 0.004).

Furthermore, it was observed that males exhibit higher mortality rated compared to females. Additionally, patients with lung cancer (38.1%) and patients with stage IV disease (82.5%) presented a higher risk of FN; however, no statistical significance was observed in these associations.

## 4. Discussion

The objective of the current study was to evaluate the incidence of FN, the duration of neutropenia, and the outcome of hospitalization due to FN in patients with solid tumors undergoing chemotherapy and immunotherapy. This prospective study involved the evaluation of 284 cancer patients who had clinically documented infections. FN episodes were identified in 63 of these patients (22.2%), which is consistent with findings in other reports [3].

Multiple risk parameters and scoring systems are employed to assess FN patients. However, several organizations, including the American Society of Clinical Oncology (ASCO), National Comprehensive Cancer Network (NCCN), Infectious Disease Society of America (IDSA), and European Society of Medical Oncology (ESMO), recommend using the Multinational Association for Supportive Care in Cancer (MASCC) score for risk stratification of patients [31,32,33]. This scoring system aims to categorize outpatients into low- and high-risk groups, thereby guiding decisions regarding hospitalization or outpatient management with oral antibiotics [28,32,33].

The risk of FN is encountered in many cancer types. Numerous risk factors contributing to FN episodes have been reported, and the risk of infection, morbidity, and mortality may vary depending on factors such as the type of solid tumor, patient characteristics, administered therapy, duration of neutropenia, site, and type of pathogen isolated [28,34,35].

It is well-documented that cancer treatment involving myelo-suppressive chemotherapy increases the susceptibility of patients to develop chemotherapy-induced neutropenia (CIN). The duration and severity of neutropenia can ultimately lead to the development of FN and its potentially life-threatening complications [36].

The administration of immunotherapy for cancer treatment can trigger immune-related adverse events (irAEs) which affect multiple organs, including the hemopoietic system, although their characterization remains limited. Delanoy et al. conducted a descriptive observational study including 745 adult patients with grade 2 or more severe hematologic irAEs induced by anti-PD-1 or anti-PD-L1. Among the patients, 35 out of 745 presented with hematologic irAEs, and 9 out of 35 (26%) developed neutropenia. Of these 9 patients, 67% developed FN and 11% succumbed to septic shock during the FN episode. Notably, in our study, most patients received chemotherapy and only one patient experiencing FN received immunotherapy. Consequently, the limited representation of patients treated with immunotherapy or biological agents does not offer a comprehensive assessment of how FN impacts cancer treatment outcomes in general; it is primarily applicable to patients undergoing chemotherapy [37].

Various studies have identified risk factors and models predicting chemotherapy-induced neutropenia and its associated complications. Lyman et al. detailed risk factors associated with FN in a systematic review, encompassing 31 studies involving patients with solid and hematologic malignancies [7]. These risk factors were categorized based on patient characteristics, treatment modalities, disease status, and genetic factors. Variables such as female gender, advanced age, poor PS (ECOG), comorbidities, and advanced disease were considered predisposing factors for FN episodes. The use of G-CSFs, which were introduced for clinical use in the 1990s, has been shown to reduce the duration and severity of neutropenia, lower FN incidence, and diminish the risk of infection and treatment-related mortality in myelo-suppressed cancer patients [36,38].

Patients receiving primary prophylaxis with G-CSFs exhibited a significant reduction in the risk of FN, estimated to be ≥20%. The type of cancer and the administration of cytotoxic chemotherapy were also implicated, although G-CSF co-administration complicated the estimation of FN risk. Another analysis conducted by Family et al., which included 16,956 patients, identified female gender, intact skin integrity, and recent corticosteroid use as significant risk factors for FN [39]. In our study, predisposing risk factors consistent with the literature were advanced disease stage, significant comorbidities, and the administration of chemotherapy. Interestingly, males were found to be at a higher risk of FN episodes than females.

As mentioned above, advanced disease is considered one of several factors, aside from chemotherapy administration, that increases the risk of FN and its associated complications. This risk escalates when one or more co-morbidities are present in the same patient [28].

In light of these considerations, the National Comprehensive Cancer Network (NCCN) has introduced guidelines that recommend prophylactic G-CSF use for patients undergoing myelo-suppressive chemotherapy who have a high risk of FN (>20%). Additionally, it suggests possible G-CSF use for patients with an intermediate risk for FN (10–20%; with ≥1 FN risk factor). These recommendations aim to reduce the incidence of FN and infection-related complications. Prophylactic G-CSF administration is also associated with sustained chemotherapy dose intensity and reduced mortality. Failure to administer G-CSF prophylaxis could be particularly detrimental in stage IV cancer patients, in contrast to those with nonmetastatic cancer [40].

The proportion of patients undergoing intense myelo-suppressive chemotherapy with curative intent has become increasingly common in patients with metastatic solid tumors and advanced NHL [40].

To mitigate the risk of FN and its associated consequences in late-stage patients, various strategies are employed. These strategies encompass adjustments in the intensity of cytotoxic treatment, the inclusion of G-CSF, and the administration of prophylactic antibiotics. Furthermore, dose reductions or treatment delays may be utilized to prevent the occurrence of FN, although it is important to note that reducing chemotherapy could potentially lead to more effective disease control and higher survival rate in malignancies with curative potential [41]. Neutrophils play a crucial role in host defense against infection, particularly those of bacterial and fungal origin. The extent and duration of neutropenia elevate the infection risk, with the highest risk observed in patients who experience profound and prolonged neutropenia following chemotherapy [30]. The correlation between the degree and duration of neutropenia and the risk of infection was initially documented in hematologic patients with acute myeloid leukemia by Bodey et al. [42]. A neutropenia duration of less than seven days categorizes patients into a lower-risk group [28].

In our study, a median neutropenia duration of three days (range 1–10) was observed. Consequently, it was opted to categorize hospitalized patients using a three-day threshold for neutropenia (<3 days or ≥3 days) in order to examine predictive factors of neutropenia duration from a unique perspective that had not been previously documented in the literature. The statistically significant results revealed a relationship between existing oropharyngeal mucositis and neutropenia for over three days. Additionally, a neutropenia duration exceeding three days was also statistically significant in terms of its impact on the overall outcome.

Carmona-Bayonas et al. conducted a study involving 1383 patients with FN episodes. Their findings indicated that an identifiable site of infection was present in 65% of cases [10]. This percentage is notably higher than the 55.6% observed in our study population. The most commonly assumed initial diagnoses in their study included upper respiratory infection (14.9%), enteritis (12.7%), stomatitis (11.8%), and acute bronchitis (10.7%) [10]. These results differ from our findings, in which urinary tract and respiratory tract infections were more prevalent, accounting for 17.5% and 14.3%, respectively. In the same study, positive cultures were detected in 20–30% of cases and positive BC in 10–25% of cases [10]. These figures are somewhat similar to our population, where 34,9% of cases with positive cultures and 17.5% with positive blood cultures were observed. The most common tumor types in their study were lung and breast cancers, constituting approximately 56% of the cases [10], whereas in our study, lung and colorectal cancers accounted for 47.6% of the cases.

Over the past decades, studies have shown a shift in the dominant pathogens responsible for neutropenic fever. While Gram-negative bacteria were previously prevalent, the increasing use of indwelling catheters, facilitating colonization by Gram-positive skin flora [9,15,43], and advancements in chemotherapy modalities [44] have led to an increase in Gram-positive pathogens [9,15,43]. Bacteremias caused by Gram-negative organisms have also risen compared to Gram-positive ones. *Enterobacteriaceae* spp. are predominant, followed by *Pseudomonas aeruginosa* and other Gram-negative pathogens. Prophylactic antibiotic use has contributed to the emergence of resistant pathogens, such as extended-spectrum beta-lactamase (ESBL)-producing *Enterobacteriaceae* and carbapenem-resistant strains.

Regarding Gram-positive bacteria, *Staphylococci* are among the most common bacteremia agents [45]. *Staphylococcus aureus*, including methicillin-resistant *strains* (MRSA), coagulase-negative staphylococci, viridans group streptococci, *Enterococcus*, and especially vancomycin-resistant *Enterococcus* (VRE), are of particular concern. Anaerobic bacteria are rare and are typically associated with polymicrobial bacteremia, especially in patients undergoing abdominal surgery [44].

In contrast to the existing literature, our study demonstrated that Gram-positive bacteria were the predominant pathogens in BC. This trend is likely attributable to the high utilization of CVC among our cohort. Specifically, *Staphylococci* were the most prevalent of the Gram-positives in patients with CVC and positive BC. Gram-negative bacteria, including *Escherichia coli*, *Enterobacter cloacae*, and *Pseudomonas aeruginosa,* were detected from positive cultures.

Fungal infections are typically considered rare, particularly in cases of shorter-duration neutropenia. However, the most significant risk factor for fungal infection is profound and prolonged neutropenia (i.e., 14 days or more with ANC < 100/μL) [44,46]. These interactions are among the leading causes of morbidity and mortality in patients with neutropenia, especially those with hematological cancer. The risk of invasive fungal infection is estimated to be 15–25% in high-risk patient groups, with over 90% of fungal infections attributed to *Candida* and *Aspergillus* species. Mortality rates are high, reaching 50% for *Candida* infections and 100% for *Aspergillus* infections [47]. The use of prophylactic fluconazole has led to an increase in non-albicans *Candida* strains. Therefore, profound and prolonged neutropenia (i.e., lasting 14 days or more with ANC < 100) remains the most significant risk factor for fungal infections [44].

In our study population, the incidence of fungal infection was notably low, with only two patients isolating fungi from their cultures. This can be attributed to a brief duration of neutropenia, lasting less than three days. In the first case, *C. parapsilosis* was isolated from urine. Simultaneously, the patient also exhibited positive blood and urine cultures for *E. cloacae*. The duration of neutropenia in this case was just one day, and the patient eventually improved. In the second case, *C. tropicalis* was detected in both blood and sputum cultures. Additionally, the patient was presented with a polymicrobial infection involving methicillin-resistant *S. hominis* from BC and pan-drug-resistant *A. baumannii* in urine culture. The duration of neutropenia in this instance extended to seven days, and the outcome was fatal.

Immunodeficiency of patients with solid tumors, in combination with FN, increases their vulnerability to life-threatening infections. Urgent administration of empirical broad-spectrum antibiotics has been a standard of therapy for nearly five decades and is proven lifesaving. The duration of treatment depends on risk assessment; low-risk neutropenic patients require shorter treatment courses, including oral regimens, in contrast to high-risk patients that may require additions and modifications of the initial regimen in combination with prolonged treatment administration. For high-risk patients, the risk for infectious complications has decreased with the selected use of prophylactic antimicrobial regimens or G-CSF [48].

FN is associated with prolonged hospitalization, substantial morbidity, and high mortality rates. As a result, it is recommended to initiate broad-spectrum empiric antimicrobial therapy promptly, even within one hour after hospital triage, due to the elevated risk of progression to sepsis and death [14,15,28]. Given that the microbial etiology is often unknown at the initiation of treatment, the choice of empiric therapy should be guided by the locally prevalent pathogens and their sensitivities, as well as the potential for complications, the site of infection, and lastly, the cost associated with various regimens [28,49].

Empiric antibiotic treatment should be initially administered and altered to appropriate specific therapy when the cause is found. Treatment is also modified depending on clinical stability or the patient’s fever pattern. In pyrexia lasting for >4–6 days, empirical initiation of antifungal therapy may be needed [28]. Consistent reassessments of hospitalized patients and therapeutic modifications according to guidelines are highly important. Patients should be discharged from the hospital as soon as they are indicated in an attempt to eliminate prolonged hospitalization that is associated with resistant pathogens and higher mortality [28,50].

None of the patients enrolled in the study received prophylactic antimicrobial therapy, and all of them received empiric antimicrobial treatment within the first hours after admission. In most cases, the initial empiric treatment was modified during their hospitalization.

It is important to note that approximately 40% to 50% of the total cost of hospitalization in cancer care is attributed to FN, which is associated with a mortality rate ranging from 3% to 20% due to various risk factors [24,35]. Among FN patients with confirmed bacteremia, the prognosis is even worse, with an 18% mortality rate in Gram-negative and 5% in Gram-positive bacteremia [28]. The mortality rates observed in our study align with those reported in the literature. However, it is important to acknowledge that the limited sample size in our study prevents us from drawing definitive conclusions about mortality.

Understanding and modulating the immune system will probably result in a lessening of treatment-induced immunodeficiency and further determine the relationship between cancer treatment, neutropenia, fever, and infection [48].

The priority concentrates on cancer treatments that result in less cytotoxicity and immunodeficiency and strategies to eliminate the adverse effects of treatment-related immunosuppression, such as empirical antibiotic regimens, prophylactic antibiotics, and hematopoietic cytokines in high-risk patients [48].

This study is subject to certain notable limitations, including the involvement of only two medical centers and a relatively small cohort. Another limitation is the fact that the vast majority of patients were chemotherapy treated, and immunotherapy was administered only to one patient.

## 5. Conclusions

FN is a prevalent and severe complication among cancer patients with solid tumors, primarily those undergoing chemotherapy. The duration of neutropenia, length of hospital stay, and patient outcomes are influenced by various risk factors. Notably, the microbiological spectrum has changed, with responsible pathogens identified in only one out of three cases. Rapid diagnosis and prompt treatment are paramount, and any delays in patient evaluation may adversely affect their prognosis.

## Figures and Tables

**Table 1 microorganisms-11-02547-t001:** Patients’ characteristics, reason for hospitalization, risk factors, and outcomes regarding the duration of neutropenia in patients with febrile neutropenia.

Patient Characteristics	All Patients *n* (%) (*n* = 63)	Patients with Duration of Neutropenia *n* (%)		*p*-Value
		<3 days (*n* = 28)	≥3 days (*n* = 35)	
**Age**	71 ± 10.2 years			
**Sex**				0.72
Male	42 (66.7)	18 (64.3)	24 (68.6)	
Female	21(33.3)	10 (35.7)	11 (31.4)	
PS (ECOG)				0.374
0–1	46 (73.0)	22 (78.6)	24 (68.6)	
2–4	17 (27)	6 (21.4)	11 (31.4)	
**Reason of hospitalization**				0.826
Chemotherapy	1 (1.6)	1 (3.6)	0	
Infection	17 (27.0)	7 (25.0)	10 (28.6)	
Disease Progression	2 (3.2)	1 (3.6)	1 (2.9)	
Fever	40 (63.5)	18 (64.3)	22 (62.9)	
Other	3 (4.8)	1 (3.6)	2 (5.7)	
**Comorbidities**				0.185
None	6 (9.5)	4 (14.3)	2 (5.7)	
Cardiovascular	25 (39.7)	14 (50.0)	11 (31.4)	
Respiratory	26 (41.3)	8 (28.6)	18 (51.4)	
Other	6 (9.5)	2 (7.1)	4 (11.4)	
**Stage**				0.47
I	2(3.2)	0 (0.0)	2 (5.7)	
II	5(7.9)	2 (7.1)	3 (8.6)	
III	4(6.3)	1 (3.6)	3 (8.6)	
IV	5282.5)	25 (89.3)	27 (77.1)	
**Disease Response**				0.72
CR, PR, SD	42 (66.7)	18 (64.3)	24 (68.6)	
PD	21 (33.3)	10 (35.7)	11 (31.4)	
Prior hospitalization (last 3 months)				0.907
No	40 (63.5)	18 (64.3)	22 (62.9)	
Yes	23 (36.5)	10 (35.7)	13 (37.1)	
**Oropharyngeal mucositis**				0.012
No	56 (88.9)	28 (100.0)	28 (80.0)	
Yes	7 (11.1)	0 (0.0)	7 (20.0)	
**Prior antimicrobial use (last 3 months)**				1
No	36 (57.1)	16 (57.1)	20 (57.1)	
Yes	27 (42.9)	12 (42.9)	15 (42.9)	
**Prior steroid use (last 1 month)**				0.37
No	56 (88.9)	26 (92.9)	30 (85.7)	
Yes	7 (11.1)	2(7.1)	5 (14.3)	
**Prior surgery (last 1 month)**				0.427
No	60 (95.2)	26 (92.9)	34 (97.1)	
Yes	3 (4.8)	2 (7.1)	1 (2.9)	
**Chemotherapy**				0.26
No	1 (1.6)	1 (3.6)	0 (0.0)	
Yes	61 (96.8)	27 (96.4)	34 (97.1)	
Immunotherapy	1 (1.6)	1 (3.6)	0 (0.0)	0.872
Chemo/immunotherapy	1 (1.6)	0 (0.0)	1 (2.9)	0.367
**Current radiotherapy (last 1 month)**				0.626
No	50 (79.4)	23 (82.1)	27 (77.1)	
Yes	13 (20.6)	5 (17.9)	8 (22.9)	
**Current invasive procedures** **(last 1 month)**				0.069
None	49 (77.8)	19 (67.9)	30 (85.7)	
CVC insertion	5 (7.9)	5 (17.9)	0 (0.0)	
Peritoneal paracentesis	1 (1.6)	0 (0.0)	1 (2.9)	
Pleural paracentesis	2 (3.2)	0 (0.0)	2 (5.7)	
Nephrostomy	1 (1.6)	0 (0.0)	1 (2.9)	
Bile stent insertion	1 (1.6)	1 (3.6)	0 (0.0)	
PTC	1 (1.6)	1 (3.6)	0 (0.0)	
Biopsy	2 (3.2)	1 (3.6)	1 (2.9)	
Other	1 (1.6)	1 (3.6)	0 (0.0)	
**G-CSF**				0.7
No	10 (15.9)	5 (17.9)	5 (14.3)	
Yes	53 (84.1)	23 (82.1)	30 (85.7)	
**Outcome**				0.05
Cure	8 (12.7)	1 (3.6)	7 (20.0)	
Improvement	47 (74.6)	25 (89.3)	22 (62.9)	
Death	8 (12.7)	2 (7.1)	6 (17.1)	

PS: Performance status, ECOG: Eastern Cooperative Oncology Group, CR: Complete response, PR: Partial response, SD: Stable disease, PD: Progressive disease, CVC: Central venous catheter, PTC: Percutaneous transhepatic cholecystectomy, G-CSF: Granulocyte colony stimulating factor.

**Table 2 microorganisms-11-02547-t002:** Neutropenia, lymphopenia, and thrombocytopenia in patients with febrile neutropenia.

Type of Cytopenia	Stratification of Cytopenia	Count of Cells	*n*	%
**Neutropenia**		Count of neutrophils/mm^3^	17	27
	Mild	1000–1500	25	39.7
	Moderate	500–1000	21	33.3
	Severe	<500	Total: 63	100
**Lymphopenia**		Count of lymphocytes/μL	13	31
	Mild	500–800	22	52.4
	Moderate	200–500	7	16.7
	Severe	<200	Total: 42	100
**Thrombocytopenia**		Count of thrombocytes × 10^3^/μL	17	37.8
	Mild	100–140	16	35.6
	Moderate	50–100	12	26.7
	Severe	<50	Total: 45	100

**Table 3 microorganisms-11-02547-t003:** Site of collected cultures in total and pathogens identified in patients with febrile neutropenia.

	Site of Collected Culture						
Pathogens	Blood	Blood through Catheter	Urine	Stool	Pus	Sputum	Total
*Staphylococcus hominis*	1	1	-	-	-	-	2
*Staphylococcus epidermidis*	-	2	-	-	-	-	2
*Staphylococcus haemolyticus*	1	-	1	-	-	-	2
*Staphylococcus aureus*	-	-	-	-	-	2	2
*Corynebacterium* spp.	-	-	-	-	-	1	1
*Enterococcus faecium*	-	1	-	1	-	-	2
*Enterococcus faecalis*	-	-	1	-	1	-	2
*Enterococcus avium*	-	-	-	-	-	1	1
*Clostiridium difficile*	-	-	-	2	-	-	2
*Micrococcus luteus*	1	-	-	-	-	-	1
*Pseudomonas aeruginosa*	1	-	-	-	-	2	3
*Escherichia Coli*	1	2	5	-	-	2	10
*Enterobacter cloacae*	1	-	3	-	-	1	5
*Serratia marcescens*	-	-	1	-	-	-	1
*Acinetobacter lwofii*	-	-	1	-	-	-	1
*Acinetobacter baumannii*	-	-	1	-	-	-	1
*Peptoniphilus asaccharolyticus*	-	-	-	-	-	1	1
**Positive Cultures Collected**	6	6	13	3	1	10	39

**Table 4 microorganisms-11-02547-t004:** Site of infection and main factors determining the outcome of hospitalization in patients with febrile neutropenia.

Factors	All Patients *n* (%)	Outcome *n* (%)			*p*-Value
		Cure	Improvement	Death	
	*n* = 63	*n* = 8	*n* = 47	*n* = 8	
**Sex**					0.715
Male	42 (66.6)	6 (75.0)	30 (63.8)	6 (75.0)	
Female	21 (33.4)	2 (25.0)	17 (36.2)	2 (25.0)	
**Site of infection**					0.347
Not found	28 (44.4)	4 (50.0)	22 (46.8)	2 (25.0)	
RTI	9 (14.3)	2 (25.0)	7 (14.9)	0 (0.0)	
UTI	11 (17.5)	1 (12.5)	7 (14.9)	3 (37.5)	
GI tract infection	2 (3.2)	1 (12.5)	1 (2.1)	0 (0.0)	
BSI	4 (6.4)	0 (0.0)	3 (6.4)	1 (12.5)	
CABSI	1 (1.6)	0 (0.0)	1 (2.1)	0 (0.0)	
Biliary infection	1 (1.6)	0 (0.0)	1 (2.1)	0 (0.0)	
Peritoneal infection	1 (1.6)	0 (0.0)	0 (0.0)	1 (12.5)	
CDI	2 (3.2)	0 (0.0)	2 (4.3)	0 (0.0)	
SSTI	3 (4.8)	0 (0.0)	2 (4.3)	1 (12.5)	
GI and CABSI	1 (1.6)	0 (0.0)	1 (2.1)	0 (0.0)	
**Comorbidities**					0.632
None	6 (9.5)	0 (0.0)	6 (12.8)	0 (0.0)	
Cardiovascular	25 (39.7)	3 (37.5)	17 (36.2)	5 (62.5)	
Respiratory	26 (41.3)	4 (50.0)	19 (40.4)	3 (37.5)	
Other	6 (9.5)	1 (12.5)	5 (10.6)	0 (0.0)	
**PS (ECOG)**					0.004
**0–1**	46 (73)	6 (75.0)	38 (80.9)	2 (25.0)	
**2–4**	17 (27)	2 (25.0)	9 (19.1)	6 (75.0)	
**Primitive location of cancer**					0.136
Lung					
NSCLC	10 (15.9)	1 (12.5)	8 (17.0)	1 (12.5)	
SCLC	14 (22.3)	2 (25.0)	11 (23.4)	1 (12.5)	
Colorectal	6 (9.6)	1 (12.5)	3 (6.4)	2 (25.0)	
Upper GI tract	4 (6.4)	1 (12.5)	3 (6.4)	0 (0.0)	
Pancreatic	5 (7.9)	1 (12.5)	4 (8.5)	0 (0.0)	
Breast	3 (4.8)	0 (0.0)	3 (6.4)	0 (0.0)	
Head and neck	3 (4.8)	0 (0.0)	2 (4.3)	1 (12.5)	
Bladder	2 (3.2)	0 (0.0)	2 (4.3)	0 (0.0)	
Kidney	1 (1.6)	1 (12.5)	0 (0.0)	0 (0.0)	
Gynecologic	5 (7.9)	0 (0.0)	4 (8.5)	1 (12.5)	
Prostate	1 (1.6)	0 (0.0)	1 (2.1)	0 (0.0)	
NET	1 (1.6)	0 (0.0)	0 (0.0)	1 (12.5)	
Sarcoma	5 (7.9)	1 (12.5)	4 (8.5)	0 (0.0)	
Occult primary	3 (4.8)	0 (0.0)	2 (4.3)	1 (12.5)	
**Stage**					0.256
I	2 (3.2)	0 (0.0)	1 (2.1)	1 (12.5)	
II	5 (7.9)	2 (25.0)	3 (6.4)	0 (0.0)	
III	4 (6.3)	0 (0.0)	4 (8.5)	0 (0.0)	
IV	52 (82.5)	6 (75.0)	39 (83.0)	7 (87.5)	
**Disease Response**					0.167
CR, PR, SD	42 (66.7)	6 (75.0)	33 (70.2)	3 (37.5)	
PD	21 (33.3)	2 (25.0)	14 (29.8)	5 (62.5)	
**Prior hospitalization**					0.686
No	40 (63,5)	5 (62.5)	31 (66.0)	4 (50.0)	
Yes	23 (36.5)	3 (37.5)	16 (34.0)	4 (50.0)	
**Oropharyngeal mucositis**					0.387
No					
Yes	56 (88.9)	6 (75.0)	43 (91.5)	7 (87.5)	
	7 (11.1)	2 (25.0)	4 (8.5)	1 (12.5)	
**Prior antimicrobial use** **(last 3 months)**					0.488
No	36 (57.1)	5 (62.5)	25 (53.2)	6 (75.0)	
Yes	27 (42.9)	3 (37.5)	22 (46.8)	2 (25.0)	
**Prior G-CSF use**					0.74
No	10 15.9	1 (12.5)	7 (14.9)	2 (25.0)	
Yes	53 84.1	7 (87.5)	40 (85.1)	6 (75.0)	
**Treatment**					
Chemotherapy	61 (96.8)	8 (100.0)	45 (95.7)	8 (100.0)	0.635
Immunotherapy	1 (1.6)	0 (0.0)	1 (2.1)	0 (0.0)	0.704
Combination	1 (1.6)	0 (0.0)	1 (2.1)	0 (0.0)	0.841

RTI: Respiratory tract infection, UTI: Urinary tract infection, GI: Gastrointestinal, BSI: Bloodstream infection, CABSI: Catheter-associated bloodstream infection, CDI: *Clostridioides* difficile infection, SSTI: Skin and soft tissue infections, PS: Performance status, ECOG: Eastern Cooperative Oncology Group, NSCLC: Non-small cell lung cancer, SCLC: Small cell lung cancer, NET: Neuroendocrine tumor, CR: Complete response, PR: Partial response, SD: Stable disease, PD: Progressive disease, G-CSF: Granulocyte colony stimulating factor.

## Data Availability

The data presented in this study are available on request from the corresponding authors.

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
