# Peer review of "Evaluation of Febrile Neutropenia in Hospitalized Patients with Neoplasia Undergoing Chemotherapy"

_microorganisms, 2023, doi:10.3390/microorganisms11102547_

Round 1
Reviewer 1 Report
The manuscript by Bachlitzanaki et al. described different aspects of hospitalized neutropenic patients with solid tumors undergoing chemotherapy or immunotherapy, afferring to the oncology department of the two largest tertiary hospitals of the island of Crete, Greece. The paper is interesting even though some points need to be better structured to allow the readers a better comprehension of the retrospective study conducted. There are areas where the paper could be improved by clarity and dept.
Abstract
- I suggest to better structure the abstract.
- It could be helpful for the reader to link the results to the conclusion. Now it is not clear why your results are important in the treatment and diagnosis of Febrile Neutropenia.
- Please, add the aim of the study in the abstract.
Introduction
- It is important to use the complete form of the acronyms when used for the first time in the main text (i.e., Blood culture line 56).
- Line 80, the example of candida must be better introduced.
Materials and Methods
- Please, make the material and methods section clearer. Dividing the section into subsection could help (i.e. patients sample, exclusion criteria, inclusion criteria, statistical analysis…).
- Lines 94-99: the sentence is too long. Please crop it in more sentences.
- Add the number of patients considered in the present study.
Results
- Lines 133-137: I would suggest moving this paragraph in the materials and methods section.
- Table 1: I suggest adding the standard deviation to the mean age of patients.
Make the table easier to read and more understandable (i.e. “reason of hospitalization” in bold).
- Line 144: “primary tumor location”. It is very confusing. Do you mean “primitive tumor localization”? I suggest rephrasing the sentence making it clearer.
- Lines 151-153: Check the numbers, I am not sure the sum of patients are correct.
- Line 156: Add “also” before radiotherapy, since, if I have understood right, they are under chemotherapy or immunotherapy and potentially under radiotherapy.
- Line 164: add “in the study” after included.
- Table 2: Reorganize the table. Add column type/stratification of the disease (mild/moderate/severe); count (ANC). It is not straightforward the understanding of “lymphopenia yes/no” “thrombocytopenia yes/no”. Please adjust the table.
- Line 224: Replace “importance” with “significance.”
Discussion
- Line 227: outcome of what? Please specify.
- Lines 275-276: please, rephrase.
- Line 303: You mention “ANC” for the first time, please write it in full form.
General consideration:
The bibliography if quite old: only the 33% (14/42) of your reference are of the last 5 years. Please add some new articles, if possible.
When you mention the name of bacteria, please, be consistent: if you use Pseudomonas aeruginosa and then you switch to P. aeruginosa it is ok but keep it that way along the text.
The English used is generally good, but I recommend a proofreading by a native English speaker to improve the clarity and readability of the paper.
Author Response
Reviewer 1
The manuscript by Bachlitzanaki et al. described different aspects of hospitalized neutropenic patients with solid tumors undergoing chemotherapy or immunotherapy, afferring to the oncology department of the two largest tertiary hospitals of the island of Crete, Greece. The paper is interesting even though some points need to be better structured to allow the readers a better comprehension of the retrospective study conducted. There are areas where the paper could be improved by clarity and dept.
Abstract
- I suggest to better structure the abstract.
Response: Thanks for the comment. Significant changes were made to the abstract, as can be seen in the revised version of the manuscript. We believe that the abstract now reads better than it did in the previous version of the manuscript.
- It could be helpful for the reader to link the results to the conclusion. Now it is not clear why your results are important in the treatment and diagnosis of Febrile Neutropenia.
Response: Thanks. The conclusions subsection of the abstract has been modified, and we fell that it better reflects the content of the abstract now.
- Please, add the aim of the study in the abstract.
Response: Thanks for the comment. We have added the aim of the study in the abstract. This can now be seen in the revised version of the manuscript.
Introduction
- It is important to use the complete form of the acronyms when used for the first time in the main text (i.e., Blood culture line 56).
Response: Thanks. We have checked the manuscript again and explained the full abbreviations in case they were not spelled out before.
- Line 80, the example of candida must be better introduced.
Response: Thanks. We have changed that part of the text, and now that part that describes the role of Candida species in candidemia is better described.
Materials and Methods
- Please, make the material and methods section clearer. Dividing the section into subsection could help (i.e. patients sample, exclusion criteria, inclusion criteria, statistical analysis…).
Response: Thanks for the comment. We have changed the methods section by introducing subsections to allow the reader to understand better the methodology we followed in the present study. It is now clearer and easier to follow. This can be seen in the methods section of the revised version of the manuscript.
- Lines 94-99: the sentence is too long. Please crop it in more sentences.
Response: Thanks. We have changed that sentence. It is now cropped in smaller sentences to be easier to read and understand as can be seen in the revised version of the manuscript.
- Add the number of patients considered in the present study.
Response: We have added that information as can be seen in the revised version of the manuscript.
- Lines 133-137: I would suggest moving this paragraph in the materials and methods section.
Response: That paragraph was removed from the results section as can be seen in the revised version of the manuscript.
- Table 1: I suggest adding the standard deviation to the mean age of patients.
Make the table easier to read and more understandable (i.e. “reason of hospitalization” in bold).
Response: Thanks. We added that information for the age of the patients. Moreover, we changed the headings of the tables. They are now in bold so that they are easier for the reader.
- Line 144: “primary tumor location”. It is very confusing. Do you mean “primitive tumor localization”? I suggest rephrasing the sentence making it clearer.
Response: Primary tumor location was rephrased to primitive tumor localization.
- Lines 151-153: Check the numbers, I am not sure the sum of patients are correct.
Response: In the text, we mention: “Concerning hospitalization etiology, fever and infection were the most common reason (63.5% and 27%, respectively), while 63.5% mentioned no previous hospitalization within the last three months.”
The numbers are correct, but we only mentioned the most common reasons for hospitalization (40 patients - 63.5% were hospitalized with fever, 17 patients - 27% with infection, 1 – 1.6% with chemotherapy, 2 – 3.2% with disease progression, and 3 – 4.8% with other reason) so that the sum is 63 patients. Considering the previous hospitalization, 40 patients had no previous hospitalization, and 23 were hospitalized within the last three months.
- Line 156: Add “also” before radiotherapy, since, if I have understood right, they are under chemotherapy or immunotherapy and potentially under radiotherapy.
Response: That is correct. The word “Also” was added before the word radiotherapy.
- Line 164: add “in the study” after included.
Response: It was added.
- Table 2: Reorganize the table. Add column type/stratification of the disease (mild/moderate/severe); count (ANC). It is not straightforward the understanding of “lymphopenia yes/no” “thrombocytopenia yes/no”. Please adjust the table.
Response: Thanks. We changed the table to conform to the reviewer's suggestions, as can be seen in the revised version of the manuscript.
- Line 224: Replace “importance” with “significance.”
Response: The word was replaced.
Discussion
- Line 227: outcome of what? Please specify.
Response: Thanks. We specified that. It is the outcome of hospitalization due to febrile neutropenia. This can be seen in the revised version of the manuscript.
- Lines 275-276: please, rephrase.
Response: Thanks. We changed that part to allow the reader to understand more easily the content as can be seen in the revised version of the manuscript. More specifically, the part “These facts that do not resemble our results, with urinary tract infections (17.5%) and respiratory tract infections (14.3%) being more common” was rephrased to “These results differ from our findings, in which urinary tract and respiratory tract infections were more prevalent, accounting for 17.5% and 14.3% respectively.”
- Line 303: You mention “ANC” for the first time, please write it in full form.
Response: ANC was first mentioned in line 111 before the revisions, and it stands for absolute neutrophil count (ANC)
General consideration:
The bibliography if quite old: only the 33% (14/42) of your reference are of the last 5 years. Please add some new articles, if possible.
Response: Thanks. We added new references that are quite more recent. The newly added articles are:
- Lehmann, D.M.; Cohen, N.; Lin, I.-H.; Alexander, S.; Kathuria, R.; Kerpelev, M.; Taur, Y.; Seo, S.K. Analyzing Adherence to the 2016 Infectious Diseases Society of America Guidelines for Candidemia in Cancer Patients. Open Forum Infect. Dis. 2022, 9, ofac555, doi:10.1093/ofid/ofac555.
- Blayney, D.W.; Schwartzberg, L. Chemotherapy-Induced Neutropenia and Emerging Agents for Prevention and Treatment: A Review. Cancer Treat. Rev. 2022, 109, 102427, doi:10.1016/j.ctrv.2022.102427.
- Delanoy, N.; Michot, J.-M.; Comont, T.; Kramkimel, N.; Lazarovici, J.; Dupont, R.; Champiat, S.; Chahine, C.; Robert, C.; Herbaux, C.; et al. Haematological Immune-Related Adverse Events Induced by Anti-PD-1 or Anti-PD-L1 Immunotherapy: A Descriptive Observational Study. Lancet Haematol. 2019, 6, e48–e57, doi:10.1016/S2352-3026(18)30175-3.
- Crawford, J.; Moore, D.C.; Morrison, V.A.; Dale, D. Use of Prophylactic Pegfilgrastim for Chemotherapy-Induced Neutropenia in the US: A Review of Adherence to Present Guidelines for Usage. Cancer Treat. Res. Commun. 2021, 29, 100466, doi:10.1016/j.ctarc.2021.100466.
- Averin, A.; Silvia, A.; Lamerato, L.; Richert-Boe, K.; Kaur, M.; Sundaresan, D.; Shah, N.; Hatfield, M.; Lawrence, T.; Lyman, G.H.; et al. Risk of Chemotherapy-Induced Febrile Neutropenia in Patients with Metastatic Cancer Not Receiving Granulocyte Colony-Stimulating Factor Prophylaxis in US Clinical Practice. Support. Care Cancer 2021, 29, 2179–2186, doi:10.1007/s00520-020-05715-3.
- Lyman, G.H. Febrile Neutropenia: An Ounce of Prevention or a Pound of Cure. J. Oncol. Pract. 2019, 15, 27–29, doi:10.1200/JOP.18.00750.
- Pizzo, P.A. Management of Patients With Fever and Neutropenia Through the Arc of Time: A Narrative Review. Ann. Intern. Med. 2019, 170, 389, doi:10.7326/M18-3192.
- Casanovas-Blanco, M.; Serrahima-Mackay, A. Febrile Neutropenia Management in Cancer Patients Receiving Anti-Cancer Agents’ Treatment: Deepening the Search to Offer the Best Care. A Critical Review Follow-Up. Crit. Rev. Oncol. Hematol. 2020, 153, 103042, doi:10.1016/j.critrevonc.2020.103042.
When you mention the name of bacteria, please, be consistent: if you use Pseudomonas aeruginosa and then you switch to P. aeruginosa it is ok but keep it that way along the text.
Response: The names of bacteria were changed throughout the text.
The English used is generally good, but I recommend a proof reading by a native English speaker to improve the clarity and readability of the paper.
Response: The manuscript was reviewed by an English speaker to make it easier for the reader.
Reviewer 2 Report
The manuscript provides an interesting evidence of the different types of infections detected in cancer patients experiencing neutropenia or febrile neutropenia.
Although the manuscript is well written and data are presented in a clear manner there are several aspects that authors need to clarify in the discussion:
1) Almost all patients analyzed were treated with chemotherapy. Only 1 received immunotherapy. Chemotherapy very often results in bone marrow toxicity and suppression. This phenomenon is very concerning for neutropenia, because the severity of neutropenia and the recovery from neutropenia using G-CSF could be long and this is due to the suppression of neutrophils progenitors in the bone marrow. The lack of patients treated with immunotherapy or biological agents in this study does not give an exhaustive picture of how the febrile neutropenia impacts cancer treatment outcome in general, but is restricted to chemo-treated patients. Authors should clarify this limitation very well in the discussion, pointing out that this trend is limited to chemo-treated patients and that can be different in patients treated with immunotherapy or other biological treatments.
In addition, since only 1 patient was treated with immunotherapy, I strongly suggests to authors to remove immunotherapy from the title
2) According with this study most of the patients that experienced febrile neutropenia were patients with late stage cancer. Late stage cancer, especially with multiple metastases can create an immune-suppressive environment and together with chemo can cause a worsening of febrile neutropenia duration or a poor outcome of infections. Authors should answer to this question in the discussion: is still appropriate to purpose chemotherapy to patients with late stage cancer due to the possibility of having deep neutropenia (especially if their bone marrow is already impaired by previous regimens of chemotherapy) and prolonged febrile neutropenia associated to infections possibly fatal? It should be considered to put them under treatments that are not so toxic for neutrophils or at least under treatments that cause transient neutropenia only attacking circulating neutrophils without attacking the bone marrow?
3) Febrile neutropenia associated to infection after cancer treatments can increase hospitalizions due to treatment of infections with antibiotics used only in hospital. This phenomenon can contribute to the increase of pathogens resistant to antibiotics, rendering the management of cancer patients more and more challenging. The increase of infections resistant to antibiotics, especially in the hospital, will increase the risk to have mortality due to pathogens resistant to antibiotics more than to the cancer progression in this patients. Authors need to discuss in an exhaustive manner this problem in the discussion, trying to give their point of view of how to mitigate this phenomenon.
Author Response
Reviewer 2
The manuscript provides an interesting evidence of the different types of infections detected in cancer patients experiencing neutropenia or febrile neutropenia.
Although the manuscript is well written and data are presented in a clear manner there are several aspects that authors need to clarify in the discussion:
1) Almost all patients analyzed were treated with chemotherapy. Only 1 received immunotherapy. Chemotherapy very often results in bone marrow toxicity and suppression. This phenomenon is very concerning for neutropenia, because the severity of neutropenia and the recovery from neutropenia using G-CSF could be long and this is due to the suppression of neutrophils progenitors in the bone marrow. The lack of patients treated with immunotherapy or biological agents in this study does not give an exhaustive picture of how the febrile neutropenia impacts cancer treatment outcome in general, but is restricted to chemo-treated patients. Authors should clarify this limitation very well in the discussion, pointing out that this trend is limited to chemo-treated patients and that can be different in patients treated with immunotherapy or other biological treatments.
Response: Thanks for the comment. We have modified the discussion section. As stated in the revised version: “It is well-documented that cancer treatment involving myelosuppressive chemotherapy increases the susceptibility of patients to develop chemotherapy-induced neutropenia (CIN). The duration and severity of neutropenia can ultimately lead to the development of FN and its potentially life-threatening complications [36].
The administration of immunotherapy for cancer treatment can trigger immune-related adverse events (irAEs) affecting multiple organs, including the hemopoietic system, although their characterization remains limited. Delanoy et al. conducted a descriptive observational study including 745 adult patients with grade 2 or more severe hematologic irAEs induced by anti-PD-1 or anti-PD-L1. Among the patients, 35 out of 745 presented with hematologic irAEs, and 9 out of 35 (26%) developed neutropenia. Of these 9 patients, 67% developed FN, and 11% succumbed to septic shock during the FN episode. Notably, in our study, most patients received chemotherapy, and only one patient experiencing FN received immunotherapy. Consequently, the limited representation of patients treated with immunotherapy or biological agents does not offer a comprehensive assessment of how FN impacts cancer treatment outcomes in general; it is primarily applicable to patients undergoing chemotherapy [37].
Various studies have identified risk factors and models predicting chemotherapy-induced neutropenia and its associated complications. Lyman et al. detailed risk factors associated with FN in a systematic review, encompassing 31 studies involving patients with solid and hematologic malignancies [7]. These risk factors were categorized based on patient characteristics, treatment modalities, disease status, and genetic factors. Variables such as female gender, advanced age, poor PS (ECOG), comorbidities, and advanced disease were considered predisposing factors for FN episodes. The use of G-CSFs, which were introduced for clinical use in the 1990s, has been shown to reduce the duration and severity of neutropenia, lower FN incidence, and diminish the risk of infection and treatment-related mortality in myelosuppressed cancer patients [36,38].”
In addition, since only 1 patient was treated with immunotherapy, I strongly suggests to authors to remove immunotherapy from the title
Response: The title was changed to “Evaluation of febrile neutropenia in hospitalized patients with neoplasia undergoing chemotherapy”
2) According with this study most of the patients that experienced febrile neutropenia were patients with late stage cancer. Late stage cancer, especially with multiple metastases can create an immune-suppressive environment and together with chemo can cause a worsening of febrile neutropenia duration or a poor outcome of infections. Authors should answer to this question in the discussion: is still appropriate to purpose chemotherapy to patients with late stage cancer due to the possibility of having deep neutropenia (especially if their bone marrow is already impaired by previous regimens of chemotherapy) and prolonged febrile neutropenia associated to infections possibly fatal? It should be considered to put them under treatments that are not so toxic for neutrophils or at least under treatments that cause transient neutropenia only attacking circulating neutrophils without attacking the bone marrow?
Response: Thanks for the interesting comment. We modified the discussion section to address that question as well. The relevant section now reads like this:
“As mentioned above, advanced disease is considered one of several factors, aside from chemotherapy administration, that increases the risk of FN and its associated complications. This risk escalates when one or more co-morbidities are present in the same patient [28].
In light of these considerations, the National Comprehensive Cancer Network (NCCN) has introduced guidelines, that recommend prophylactic G-CSF use for patients undergoing myelosuppressive chemotherapy who have a high risk for FN (> 20%). Additionally, it suggests possible G-CSF use for patients with an intermediate risk for FN (10–20%; with ≥ 1 FN risk factor). These recommendations aim to reduce the incidence of FN and infection-related complications. Prophylactic G-CSF administration is also associated with sustained chemotherapy dose intensity and reduced mortality. Failure to administer G-CSF prophylaxis could be particularly detrimental in stage IV cancer patients, in contrast to those with nonmetastatic cancer [40].
The proportion of patients undergoing intense myelosuppressive chemotherapy with curative intent has become increasingly common in patients with metastatic solid tumors and advanced NHL [40].
In an effort to mitigate the risk of FN and its associated consequences in late stage patients, various strategies are employed. These strategies encompass adjustments in the intensity of cytotoxic treatment, the inclusion of G-CSF, and the administration of prophylactic antibiotics. Furthermore, dose reductions or treatment delays may be utilized to prevent the occurrence of FN, although it is important to note that reducing chemotherapy could potentially lead to more effective disease control and higher survival rate in malignancies with curative potential [41].”
3) Febrile neutropenia associated to infection after cancer treatments can increase hospitalizations due to treatment of infections with antibiotics used only in hospital. This phenomenon can contribute to the increase of pathogens resistant to antibiotics, rendering the management of cancer patients more and more challenging. The increase of infections resistant to antibiotics, especially in the hospital, will increase the risk to have mortality due to pathogens resistant to antibiotics more than to the cancer progression in this patients. Authors need to discuss in an exhaustive manner this problem in the discussion, trying to give their point of view of how to mitigate this phenomenon.
Response: Thanks. We modified the discussion section to address this comment as well. It now reads like this:
“Immunodeficiency of patients with solid tumors in combination to FN, increases their vulnerability to life threatening infections. Urgent administration of empirical broad-spectrum antibiotics has been a standard of therapy for nearly five decades and is proven lifesaving. Duration of treatment depends on risk assessment; low-risk neutropenic patients require shorter treatment course, including oral regimens, in contrast to high-risk patients that may result in additions and modifications of the initial regimen, in combination with prolonged treatment administration. For high-risk patients, the risk for infectious complications has decreased with the selected use of prophylactic antimicrobial regimens or G-CSF [48].
FN is associated with prolonged hospitalization, substantial morbidity and high mortality rates. As a result, it is recommended to initiate broad-spectrum empiric antimicrobial therapy promptly, even within one hour after hospital triage, due to elevated risk of progression to sepsis and death [14,15,28]. Given that the microbial etiology is often unknown at the initiation of treatment, the choice of empiric therapy should be guided by the locally prevalent pathogens and their sensitivities, as well as the potential for complications, the site of infection, and lastly, the cost associated with various regimens [28,49].
Empiric antibiotic treatment should be initially administered, and altered to appropriate specific therapy when the cause is found. Treatment is also modified depending on clinical stability or patients fever pattern. In pyrexia lasting for >4–6 days, empirical initiation of antifungal therapy may be needed [28]. Consistent reassessment of hospitalized patients, and therapeutic modifications according to guidelines are of high importance. Patients should be discharged from hospital as soon as they are indicated, in an attempt to eliminate prolonged hospitalization that is associated with resistant pathogens and higher mortality [28,50].”
Round 2
Reviewer 1 Report
Thank you very much for your work. I believe that in the present form the work is acceptable for publication.